# Evaluation of a Project Integrating Financial Incentives into a Hepatitis C Testing and Treatment Model of Care at a Sexual Health Service in Cairns, Australia, 2020–2021

**DOI:** 10.3390/v16050800

**Published:** 2024-05-17

**Authors:** Joshua Dawe, Carla Gorton, Rhondda Lewis, Jacqueline A. Richmond, Anna L. Wilkinson, Alisa Pedrana, Mark Stoové, Joseph S. Doyle, Darren Russell

**Affiliations:** 1Disease Elimination, Burnet Institute, Melbourne 3004, Australia; 2Cairns Sexual Health Service, Cairns 4870, Australiadarren.russell@health.qld.gov.au (D.R.); 3School of Public Health and Preventive Medicine, Monash University, Melbourne 3004, Australia; 4Melbourne School of Population and Global health, University of Melbourne, Melbourne 3010, Australia; 5Australian Research Centre in Sex, Health and Society, La Trobe University, Melbourne 3086, Australia; 6Department of Infectious Diseases, Alfred Health and Monash University, Melbourne 3004, Australia; 7School of Medicine and Dentistry, James Cook University, Townsville 4814, Australia

**Keywords:** hepatitis C, people who inject drugs, primary care, financial incentives

## Abstract

Background: Understanding the effectiveness of novel models of care in community-based settings is critical to achieving hepatitis C elimination. We conducted an evaluation of a hepatitis C model of care with financial incentives that aimed to improve engagement across the hepatitis C cascade of care at a sexual health service in Cairns, Australia. Methods: Between March 2020 and May 2021, financial incentives were embedded into an established person-centred hepatitis C model of care at Cairns Sexual Health Service. Clients of the Service who self-reported experiences of injecting drugs were offered an AUD 20 cash incentive for hepatitis C testing, treatment initiation, treatment completion, and test for cure. Descriptive statistics were used to describe retention in hepatitis C care in the incentivised model. They were compared to the standard of care offered in the 11 months prior to intervention. Results: A total of 121 clients received financial incentives for hepatitis C testing (antibody or RNA). Twenty-eight clients were hepatitis C RNA positive, of whom 92% (24/28) commenced treatment, 75% (21/28) completed treatment, and 68% (19/28) achieved a sustained virological response (SVR). There were improvements in the proportion of clients diagnosed with hepatitis C who commenced treatment (86% vs. 75%), completed treatment (75% vs. 40%), and achieved SVR (68% vs. 17%) compared to the pre-intervention comparison period. Conclusions: In this study, financial incentives improved engagement and retention in hepatitis C care for people who inject drugs in a model of care that incorporated a person-centred and flexible approach.

## 1. Introduction

Following the introduction of unrestricted access to direct acting antiviral (DAA) therapy for the treatment of hepatitis C in March 2016, Australia made substantial early advancements towards hepatitis C elimination [1,2]. However, DAA prescribing has steadily decreased since, and at the end of 2020 an estimated 118,000 people were still living with chronic hepatitis C [3]. Therefore, further efforts are required to engage the remaining people living with hepatitis C to achieve elimination in Australia [3,4,5].

People living with and at-risk of hepatitis C experience numerous social and structural barriers that delay or prevent engagement with health services and access to hepatitis C care [6,7]. These barriers include gaps in knowledge of hepatitis C testing and treatment options among people living with hepatitis C [8] and healthcare professionals [6], the need to balance competing health, social, and financial priorities, and experiences of stigma and discrimination [9,10,11]. Consequently, it is critical that hepatitis C models of care employ a person-centred and flexible approach that accommodates the unique experiences and circumstances of each person accessing care.

Financial incentives have been proposed as an intervention to overcome barriers to hepatitis C care, particularly economic and financial barriers. Financial incentives have been used to improve hepatitis C testing and treatment uptake across multiple settings, including primary, community, and tertiary care [12,13,14,15,16]. However, while there are promising signs for the effectiveness of financial incentives, much of the existing research describes interventions not specifically tailored to people who inject drugs [13,17,18,19,20]. Consequently, there is a need for further research evaluating the effectiveness of financial incentives in hepatitis C care, particularly among people who inject drugs [21].

The primary aim of our study was to evaluate whether embedding financial incentives and strengthening referral pathways in a person-centred model of care at an Australian sexual health service was effective at facilitating engagement and retention in hepatitis C testing and treatment among people who inject drugs. Our secondary aim was to understand the experiences of implementing a financial incentive model among project staff.

## 2. Methods

### 2.1. Study Setting and Context

In 2016, an estimated 3854 people in the Cairns community (population 150,000) were living with hepatitis C, corresponding to a prevalence of approximately 2.5%, almost twice the estimated Australian prevalence of 0.94% [22]. Recognising the need for a ‘micro-elimination’ response (defined as viral elimination within a defined population), and buoyed by the widespread availability of DAA therapy, the Cairns Sexual Health Service (‘the Service’) established a person-centred hepatitis C model of care [23]. In the first 24 months of implementing the model of care, over 1000 people in the Cairns community were treated for hepatitis C [24]. By 2019, the Cairns region had made significant advancements towards hepatitis C micro-elimination, with declines in the number of newly acquired hepatitis C notifications [22,25]. However, despite ongoing efforts, many people with hepatitis C living in Cairns remained unengaged in care. Consequently, the healthcare team at the Service embedded cash-based financial incentives to enhance their usual standard of care and improve engagement and retention in hepatitis C care.

### 2.2. Evaluation Methods

We evaluated an implementation project that used financial incentives to support hepatitis C testing and treatment uptake at a sexual health service. The evaluation was conducted according to the Standards for Quality Improvement Reporting Excellence (SQUIRE 2.0) guidelines [26].

### 2.3. Hepatitis C Model of Care

Cash incentives were embedded within an existing person-centred hepatitis C model of care, which was designed to support clients through the hepatitis C cascade of care. Clients attending the Service with no previously recorded positive hepatitis C test (either antibody or RNA) were offered a hepatitis C test via standard venipuncture. A hepatitis C antibody test was performed for clients whose hepatitis C status was unknown. A hepatitis C RNA test was performed for clients who self-reported a previous positive hepatitis C test or had a previously recorded positive hepatitis C test. To reduce the need for multiple venipunctures from clients, reflexive testing was ordered for positive hepatitis C antibody tests. Clients whose hepatitis C tests (antibody and RNA) were non-reactive were informed of their results via text. Clients whose hepatitis C RNA tests were positive were contacted via phone and invited to return to the Service to discuss their results, receive a liver stiffness measurement via transient elastography, and receive a prescription for DAA therapy. The cost of DAA treatment prescription co-payment for clients on a health care card (AUD 6.60 and AUD 42.50, depending on the client’s eligibility for concession) was reimbursed to reduce the client’s out-of-pocket costs. 

### 2.4. Intervention Design and Recruitment

A project with embedded cash incentives in the existing hepatitis C model of care was implemented through the Service between 9 March 2020 and 12 May 2021. Any client who self-reported drug use through injection was eligible to participate. No appointment or referral was required for participation. 

A total of AUD 6000 in funding was allocated for financial incentives, which was offered to eligible clients until the funding pool was exhausted. Clients were offered a cash incentive by clinical staff (including nurses, GPs, and sexual health physicians) of the Service across the hepatitis C cascade of care (hepatitis C testing, returning for positive results, treatment commencement, treatment completion, SVR test for cure) to support their engagement in hepatitis C testing and treatment (Figure 1). Collection of the first box of DAA treatment indicated treatment commencement, and collection of the final box of DAA treatment indicated treatment completion. 

The project began offering a AUD 10 cash incentive in March 2020. The financial incentive amount was determined based on the expected number of incentive payments and the size of the budget. To reflect changes in the financial circumstances of clients during the initial COVID-19 outbreak, cash incentives were increased to AUD 20 from June 2020 until the project ended in May 2021. All clients who received a hepatitis C test were offered an additional AUD 20 financial incentive to refer a peer from their injecting network. The incentive money was stored in the pharmacy of the Service, and signatures from two staff members were required each time the cash was accessed. In addition to the availability of financial incentives, there were variations in service delivery during the intervention, including the use of promotional posters, the establishment of a peer referral pathway, and increased efforts to respond to the needs of clients by storing medication onsite or delivering it to them in the community.

The project utilised several strategies for recruitment. Firstly, clients attending the Service at any time during clinic operating hours who self-reported drug use through injection to a clinician were offered a financial incentive to receive a hepatitis C test by service staff. Promotional posters advertising the financial incentives for hepatitis C testing were displayed within the Service to enhance this recruitment pathway. Secondly, the incentives project was promoted by Service staff during regular outreach activities at nearby health and harm reduction services, including needle and syringe programs (NSPs) and homelessness services. Thirdly, nearby health and harm reduction services, which provide tailored services for people who inject drugs, were informed of the financial incentives project and encouraged to refer clients to the Service. These health services included NSPs, mental health services, and drug and alcohol treatment services. Lastly, clients who received a financial incentive for hepatitis C testing were offered an additional cash incentive to refer peers from within their injecting network, with a maximum of three referrals per client. Peer referrals were temporarily paused between March 2020 and May 2020 in response to COVID-19 restrictions, and recommenced in June 2020 when restrictions were eased.

### 2.5. Data Collection

#### 2.5.1. Quantitative Data

All clients of the Service who received a financial incentive for hepatitis C testing were included in the analysis. Clinical and demographic data were available from the electronic medical record system of the Service. Financial incentives data were collected prospectively by project staff using a purpose-built Microsoft Excel worksheet.

To compare the hepatitis C cascade of care before and during the implementation of financial incentives, we analysed aggregated routinely collected Quality Assurance data of hepatitis C treatment outcomes (DAA treatment initiation, DAA treatment completion, and sustained virological response (SVR)) among clients who received a positive hepatitis C RNA test between 1 April 2019 and 1 March 2020. 

Demographic characteristics of clients included age, sex, and whether clients identified as Aboriginal and/or Torres Strait Islander. The pathway through which clients attended the Service was also recorded (self-referral, peer referral, outreach activities, referred by a nearby health service). Health services included General Practitioners, needle and syringe programs, mental health services, and drug and alcohol treatment services. Additional information included whether clients referred a peer to the Service for hepatitis C testing or if they had previously attended the Service.

#### 2.5.2. Conversations with Service Staff

To address the aim of understanding learnings from project staff when conducting a financial incentive model, we conducted interviews with four staff members from the Service involved in implementing the financial incentives program. Participants were interviewed via a semi-structured interview question guide developed by the authors to explore the experiences and learnings of staff in implementing the financial incentives project, including why they believed the incentive was necessary to enhance hepatitis C care at the Service, the facilitators, challenges, and successes of the project, and what professional learnings they gained throughout the project. Project staff were sent a participant information form before each interview, and consent was provided at the start of each interview. The interviews were conducted by the first author (JDa) via video conference.

### 2.6. Data Analysis

To evaluate the effectiveness of the financial incentives model to retain clients in hepatitis C treatment, descriptive statistics of clinical data were used to analyse key steps in the hepatitis C cascade of care, including hepatitis C antibody and RNA testing, diagnosis, treatment, and SVR12. Collection of the first box of DAA treatment from the Service pharmacy indicated treatment commencement, and collection of the final box of DAA treatment from the Service pharmacy indicated treatment completion.

A hepatitis C cascade of care was used to compare treatment uptake and retention while implementing the financial incentives model and comparison period prior to the intervention. The number and proportion of clients diagnosed with chronic hepatitis C who started and were retained in hepatitis C care (DAA treatment initiation, DAA treatment completion, and SVR) was presented. Agresti–Coull 95% confidence intervals were calculated for proportions. Data analysis was conducted using Stata Version 17.0 [27].

The initial coding framework was developed deductively based on the semi-structured interview question guide and reviewed to identify emerging themes using an inductive approach. The coding framework was created by the first author (JDa), and interviews were independently cross-checked by another author (JR) to ensure appropriate and reliable coding. The coding frameworks were developed using NVIVO-12.

### 2.7. Ethics

Evaluation of the Cairns Sexual Health Service financial incentives project received approval from the Far North Queensland Human Research Ethics Committee (Project ID 77663). This evaluation utilised Quality Assurance data routinely collected during clinical care at the Service to describe the hepatitis C cascade of care; therefore, a waiver of consent was granted to analyse Quality Assurance data. Ethics approval to transcribe and analyse conversations with project staff was also obtained.

## 3. Results

A total of 121 clients received a financial incentive for hepatitis C testing between 9 March 2020 and 12 May 2021 at the Service (Table 1). Most clients were male (67%), the average age was 40 years (SD 12.5, range 18–67), and approximately one in three identified themselves as Aboriginal and/or Torres Strait Islander (26%). One in four clients tested were referred to the Service by a peer (23%). Of the 121 people who were tested for hepatitis C and received a financial incentive, 58 (48%) were tested at their first clinical consultation at the Service. Among the 28 clients who tested hepatitis C RNA positive, most were male (75%) and did not identify as Aboriginal and/or Torres Strait Islander (82%). Around half were aged between 35 and 49 years (43%) and had not previously attended the Service (50%).

### 3.1. Hepatitis C Testing and Treatment Outcomes

Among the 121 clients who received a financial incentive for a hepatitis C test, 84 (69%; 95% CI: 60–77%) were hepatitis C antibody positive (Figure 2). Among the 84 clients who were hepatitis C antibody positive, 54 (64%; 95% CI: 90–99%) were known (self-reported or Service clinical record) to be previously hepatitis C antibody positive and were hepatitis C RNA tested upon referral. Of the 84 clients who were hepatitis C antibody positive and eligible for hepatitis C RNA testing, three were lost to follow-up and did not receive a hepatitis C RNA test. Among the 81 clients who received an RNA test, 28 (35%; 95% CI: 24–46%) were RNA positive. Of these, 26 (93%; 95% CI: 76–99%) clients returned to the Service to receive their results, of whom 24 (92%; 95% CI: 75–99%) received a DAA prescription and commenced treatment. Among the 24 clients who started treatment, 21 of 24 (88%; 95% CI: 67–97%) completed treatment, and 19 of 21 (90%; 95% CI: 70–99%) obtained a SVR test 12 weeks post-treatment completion (Figure 2). All 19 clients who received an SVR test attained viral clearance. 

Across the 11-month pre-intervention period, 72% (13/18; 95% CI: 47–90%) of clients who were diagnosed with hepatitis C commenced treatment at the Service, 28% (6/18; 95% CI: 10–53%) completed treatment at the Service, and 6% (1/18; 95% CI: 1–27%) received a test for cure at the Service. During the intervention period, 86% (24/28; 95% CI: 67–96%) of clients diagnosed with hepatitis C commenced treatment, 75% (21/28; 95% CI: 55–89%) completed treatment, and 68% (19/28; 95% CI: 48–84%) received a test for cure (Figure 3). There was overlap in the 95% confidence intervals for the proportion of clients who started treatment between the intervention and comparison periods, and no overlap for the proportion of clients who completed treatment and those who received a test for cure.

### 3.2. Reflections from Project Staff

Semi-structured interviews with four staff at the Service were conducted between September and November 2022. The duration of the interviews was, on average, 47 min (range 40 to 60 min). Conversations with project staff were coded into three main themes:Financial incentives provided direct benefits to clients and opportunities for peer referrals;Financial incentives must be combined with a person-centred approach to overcome barriers to accessing hepatitis C care;Promoting the incentives program through local health and harm reduction services was critical.

### 3.3. Financial Incentives Provided Auxiliary Benefits to Clients and Opportunities for Peer Referrals 

Staff members reflected on the benefits that the financial incentives provided to clients. Engaging clients through a peer referral pathway was considered crucial to the success of financial incentives. The financial incentives also had direct benefits for clients experiencing socioeconomic disadvantage by supporting them to address immediate challenges, such as affording food and accommodation. Staff described the important role a positive healthcare experience had in supporting and encouraging clients to address other health needs and refer peers from within their networks. For example, staff reported that several clients received STI testing (syphilis, chlamydia, gonorrhoea) and were provided with take-home Naloxone. Furthermore, the peer referral process meant that people who were not clients of the Service became aware of the project through ‘word of mouth’. 


*“[Incentives] have led to people being willing to come forward … being tested and trying to have their other health care needs met in some way.”*
(Interviewee 1)


*“Suddenly, they’ve got money for food or for rent that week that they may not have had otherwise. So those more important things are taken care of, and we provide free hep C medication and try and make things as easy as possible.”*
(Interviewee 4)


*“So, word of mouth and their social groups where they get a lot of their information has that rippling effect… you reach a point where a lot of the people in that network are comfortable with what we’re doing with our service, and it suddenly becomes a lot easier to engage people.”*
(Interviewee 3)

### 3.4. Financial Incentives Must Be Combined with a Person-Centred Approach to Overcome Barriers to Accessing Hepatitis C Care 

Staff members described the importance of ensuring a person-centred approach when engaging with clients and offering financial incentives. Whilst all four staff members had positive reflections on the use of financial incentives, they also highlighted that many clients had previously experienced stigma and discrimination in health services and required respectful, non-judgmental healthcare to overcome these experiences. Being responsive and flexible to the needs of clients was described as fundamental to the success of the financial incentives model. Examples of the flexible, person-centred approach used by staff included storing medication on-site for clients experiencing unstable housing, allowing walk-in clients to access hepatitis C testing without a referral or booking, and organising for medication to be delivered to clients’ homes or nearby health services more accessible to them.


*“As well as the money, it’s that our team are actually engaging with people, and it might be the first time in a long time that they’ve had a positive engagement with the health system... some … have had rotten time at emergency departments, outpatient clinics, general practices, and are really suspicious and don’t like … health care practitioners. At our service, they’re important to us, we want to treat them.*
(Interviewee 4)


*You’re still going to need to support people. We need incentives, but we also need people to run around and meet chaps down at the shopping centre, and the ones with cognitive functioning issues or mental health issues or homelessness.”*
(Interviewee 2)

### 3.5. Promoting the Incentives Program through Local Health and Harm Reduction Services Was Critical

Staff reflected that the incentive program provided an opportunity to strengthen existing partnerships with nearby health and harm reduction services, including NSPs, mental health services, and homelessness services to broaden the reach of their hepatitis C program using flyers and posters. Partner services also shared information about the financial incentives project with their existing clients and encouraged them to participate in the financial incentives project, which resulted in new clients engaging in hepatitis C care both at the Service and during outreach activities. Through this approach, new referral pathways were established from trusted and familiar healthcare services for people who inject drugs to the Service. 

Clients were more likely to consider the Service as a safe space to receive hepatitis C care. See the previous comments in the Results. The referral and, therefore, an implicit recommendation to the Service from peers, some of whom had already presented and received HCV care, sends a strong message about the service being a “safe” space.


*“I think another enabler was just multiple sites of access and promotion around hepatitis C. We worked with sexual health, GPs, the needle and syringe program, drug and alcohol, homelessness services, youth services. We took the hep C message to a whole range of services that were able to have the conversation and refer people for testing and treatment.”*
(Interviewee 1)

## 4. Discussion

In this study, we found that integrating financial incentives into a person-centred hepatitis C model of care is an effective strategy for reaching new clients and re-engaging with existing clients in hepatitis C care. Among the clients diagnosed with hepatitis C during the intervention, almost all received a prescription for DAA therapy, and three in four completed DAA treatment. In the context of mathematical modelling from Australia demonstrating that financial incentives result in modest increases in treatment initiation among people diagnosed with chronic hepatitis C and remain cost-effective up to AUD 200, the financial incentives model evaluated in this study was also likely highly cost-effective [28].

There is currently limited evidence for increased retention across the hepatitis C cascade of care among people who inject drugs when financial incentives are offered, with previous studies primarily using non-comparative designs [14,16,29,30]. In this study, we compared treatment retention among clients diagnosed with chronic hepatitis C during project implementation to 11 months prior to the intervention. Among clients with a positive hepatitis C RNA test, there were improvements in the proportion who initiated treatment (93% vs. 72%), completed treatment (75% vs. 28%), and achieved cure (68% vs. 6%). Importantly, the intervention of financial incentives was strengthened with support mechanisms to improve care retention, such as health promotion activities, peer referrals, and strengthening referral pathways to nearby health services. Given this was not a controlled trial, these factors likely contributed to improvements in hepatitis C treatment retention during the project. Nonetheless, our observation aligns with previous studies, which found that providing financial incentives was associated with improvements linked to hepatitis C care and treatment initiation among people who inject drugs [12,31]. 

Another notable finding was that one in four clients were directly referred to the Service by individuals within their peer network, and one in two clients had never previously received healthcare through the Service. This finding suggests that financial incentives are an effective strategy for engaging new clients in hepatitis C testing and treatment. Furthermore, among clients who received financial incentives for hepatitis C testing during the project, one in five referred a peer to the Service, underscoring the importance of fostering relationships with people who inject drugs when implementing hepatitis C interventions, whatever their design. This observation was further reflected during conversations with project staff, who commented on the importance of engaging and developing trust within the community. These results emphasise the important role peers play in promoting financial incentives, spreading information within networks, and facilitating access to hepatitis C care in established primary and community settings that function within an often unfamiliar and forbidding health system [32].

Conversations with project staff revealed influential factors in the success of the project, and offered insights into the benefits of providing financial incentives. Project staff recognised that many clients of the Service had previous experiences of stigma and discrimination in healthcare settings, and emphasised the importance of employing a flexible, non-judgmental, and person-centred approach. This approach included enhanced DAA treatment support, such as delivering medication to clients in the community who reported difficulties accessing their prescription or returning to the Service. Project staff also highlighted the direct benefits financial incentives offer to clients. Many of the clients who received financial incentives were experiencing socioeconomic disadvantage, and financial incentives helped address financial challenges. 

## 5. Limitations

This evaluation had limitations. First, although we presented historic clinical data to allow comparison between hepatitis C testing and treatment outcomes before and during the implementation of financial incentives, our study did not have a parallel control arm, which hindered our ability to draw conclusions about the direct effectiveness of financial incentives. Second, semi-structured interviews were only conducted with four project staff involved in the implementation of the project and did not include any clients of the Service. Consequently, the views and experiences shared and the findings from our evaluation may not reflect the views of clients who received incentives for hepatitis C testing and treatment. Third, our comparator, which evaluated improvements in retention for hepatitis C treatment during the intervention period, did not account for any changes in clinic practice or over time. Fourth, our evaluation had no data indicating the number of clients offered hepatitis C testing during the intervention or before. Therefore, we could not assess the uptake of hepatitis C testing. Fifth, the project was implemented within a single sexual health service. Therefore, the findings of this study may apply to services in other geographical and health settings. Sixth, given that the interviews with project staff constituted a small, complementary component of our study, one author was considered sufficient to develop the coding framework with a second author to review it. However, we acknowledge that there are more comprehensive approaches to qualitative data analysis. Lastly, the data we used to describe the testing and treatment outcomes were obtained from the electronic medical record system of the Service. Therefore, clients who received further care in other settings not known to the Service would be incorrectly recorded as untreated. 

## 6. Conclusions

The results of our evaluation provide important evidence that incentives are an effective mechanism for engaging and retaining people in hepatitis C care when combined with a flexible, person-centred approach. Eliminating hepatitis C as a public health threat will only be achieved through increased accessibility and availability of hepatitis C care pathways in community and primary care settings, both in Australia and globally. Our findings suggest that financial incentives should be considered in settings that have experienced reductions in hepatitis C testing and treatment. However, as with all hepatitis C interventions, effective implementation is contingent on embedding financial incentives within flexible and person-centred models of care.

## Figures and Tables

**Figure 1 viruses-16-00800-f001:**
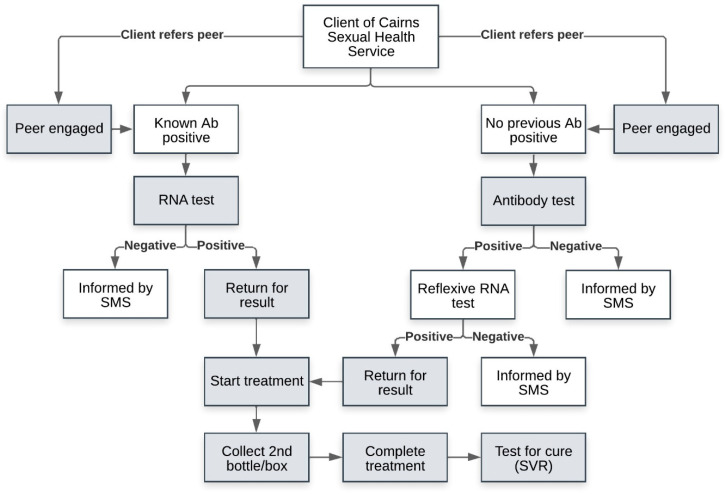
Flow diagram describing Cairns Sexual Health Service model of care with incentive payments ^a^. ^a^ Shaded boxes represent the points at which financial incentives were offered.

**Figure 2 viruses-16-00800-f002:**
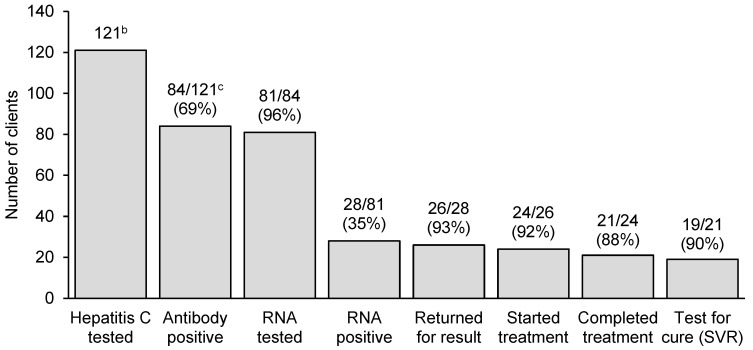
Hepatitis C cascade of care among people who received a financial incentive for hepatitis C testing at the Cairns Sexual Health Service ^a^, (N = 121). ^a^ Number of people in the previous stage of cascade of care is denominator in percentages. ^b^ Number of clients who received a hepatitis C test (antibody or RNA). ^c^ Fifty-four clients were known to be hepatitis C antibody positive and initially received a hepatitis C RNA test.

**Figure 3 viruses-16-00800-f003:**
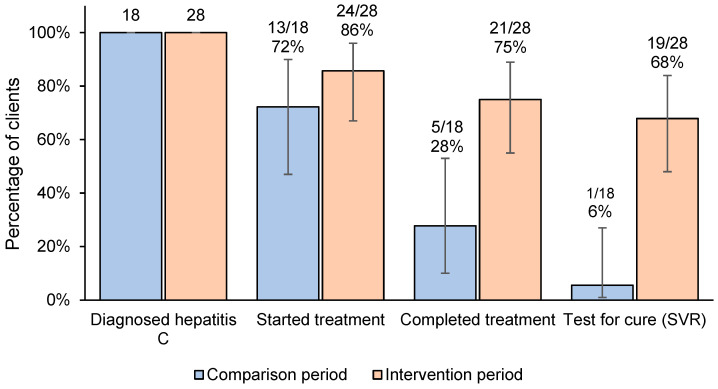
Comparison of hepatitis C treatment outcomes among clients who received a positive hepatitis C RNA test before and after the introduction of financial incentives ^a,b^, (n = 46). ^a^ Number of clients diagnosed with chronic hepatitis C is denominator in percentages. ^b^ Bars represent 95% confidence intervals for proportions.

**Table 1 viruses-16-00800-t001:** Characteristics of clients who received financial incentives for hepatitis C tests, N = 121 ^a^.

Characteristic, n (%) ^b^		RNA Positive, n (%) ^b^
Total	121	28
Sex		
Male	81 (67)	21 (75)
Female	40 (33)	7 (25)
Age group (years)		
18–34	42 (35)	7 (25)
35–49	53 (44)	12 (43)
50+	26 (21)	9 (32)
Aboriginal or Torres Strait Islander		
Yes	32 (26)	5 (18)
No	89 (74)	23 (82)
Referral pathway		
Self-referral	41 (34)	7 (25)
Peer referral	28 (23)	9 (32)
Outreach activities	21 (17)	8 (29)
Referred by nearby health service ^c^	31 (26)	7 (25)
Referred another peer		
Yes	23 (19)	-
No	98 (81)	-
Previously recorded visit with CSHS		
Yes	58 (48)	14 (50)
No	63 (52)	14 (50)

NSP: Needle and syringe programme; CSHS: Cairns Sexual Health Service. ^a^ Numbers are suppressed where a cell contains less than five clients. ^b^ Percent is column percentage. ^c^ Nearby health services included General Practitioners, needle and syringe programs, mental health services, and drug and alcohol treatment services.

## Data Availability

Data are contained within the article.

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
