# Peer review of "Evaluation of a Project Integrating Financial Incentives into a Hepatitis C Testing and Treatment Model of Care at a Sexual Health Service in Cairns, Australia, 2020–2021"

_viruses, 2024, doi:10.3390/v16050800_

Round 1

Reviewer 1 Report

Comments and Suggestions for Authors

The authors in their manuscript titled ‘Evaluation of a project integrating financial incentives into a hepatitis C testing and treatment model of care at a sexual health service in Cairns, Australia present a descriptive analysis of patients who received incentives for hepatitis C testing and treatment at a sexual health service in Australia. The authors numerically compare proportion of program participants who achieve each step in the HCV care continuum to participants who received services without financial incentives in that the same location in the 11 months prior. The paper describes an approach to real world implementation of financial incentives in which participants received a financial incentive for completing each step of the HCV care cascade (testing, return for positive results, treatment commencement, treatment completion and SVR) and capture rates of progress across each step of the continuum.  

Major comments

How was the amount of the financial incentive determined? This is important information for settings that may want to implement similar projects

Qualitative interviews were conducted with 4 participants with coding done by one author and crosschecked by another author. These are not rigorous qualitative analysis methods. The authors state that interviews were conducted with staff members to explore learning. No information is provided about the type of learning being sought, the role/types of  staff interviewed,  if an interview guide was used to conduct interviews, if there was a framework that guided development of the guide ( if one was used) or if thematic saturation was sought in the qualitative analysis. The lack of these elements makes the qualitative findings difficult to interpret

Page 5 line 194- It is unclear how NVIVO was used to develop coding frameworks or what the frameworks being referred to are. Were transcriptions of interviews coded in NVIVO?

Page 5 line 212-84 were antibody positive but 54 were known antibody positive and received RNA testing. Based on study methods section, those individuals would not have been antibody tested? Were all 84-antibody tested through the study?

The choice of an 11-month period pre intervention and a comparison period seems odd. Why not a 1-year period pre intervention? Please provide the rationale for the choice of 11 months

The fact that this was a single site study in one setting should also be listed as a limitation as this may not reflect the experience in other settings

Author Response

Reviewer One

The authors in their manuscript titled ‘Evaluation of a project integrating financial incentives into a hepatitis C testing and treatment model of care at a sexual health service in Cairns, Australia present a descriptive analysis of patients who received incentives for hepatitis C testing and treatment at a sexual health service in Australia. The authors numerically compare proportion of program participants who achieve each step in the HCV care continuum to participants who received services without financial incentives in that the same location in the 11 months prior. The paper describes an approach to real world implementation of financial incentives in which participants received a financial incentive for completing each step of the HCV care cascade (testing, return for positive results, treatment commencement, treatment completion and SVR) and capture rates of progress across each step of the continuum. 

Specific comments:

  1. Reviewer: How was the amount of the financial incentive determined? This is important information for settings that may want to implement similar projects

Response: A total of AU$6,000 in funding was allocated for financial incentives, and was offered to eligible clients until the funding pool was exhausted. The budget for financial incentives was pre-determined prior to the project. Initially, project staff estimated that incentive payments of AU$10 would be sufficient to engage clients. However, this was subsequently increased to AU$20. We have updated the interview design and recruitment section to clarify this, and it now includes the following text:

“The financial incentive amount was determined on the basis of the expected number of incentives payments that would be made, and the size of the budget.” [Lines 128–130, Methods]

  1. Reviewer: Qualitative interviews were conducted with 4 participants with coding done by one author and crosschecked by another author. These are not rigorous qualitative analysis methods. The authors state that interviews were conducted with staff members to explore learning. No information is provided about the type of learning being sought, the role/types of staff interviewed,  if an interview guide was used to conduct interviews, if there was a framework that guided development of the guide ( if one was used) or if thematic saturation was sought in the qualitative analysis. The lack of these elements makes the qualitative findings difficult to interpret

Response: We have updated the limitations in response to the reviewer’s comment that our coding approach was not rigorous qualitative analysis methods with the following text:

“Sixth, given that the interviews with project staff constituted a smaller, complementary component of our study, it was considered sufficient for one author to develop the coding framework, and it be reviewed by a second author. However, we acknowledge that there are more comprehensive approaches to qualitative data analysis.” [Lines 377–381, Limitations]

We have also updated the methods section to now include more information on the design and aims of the semi-structured interview question guide:

“Participants were interviewed via a semi-structured interview question guide, which was developed by the authors to explore the experiences and learnings of staff in implementing the financial incentives project, including why they believed the incentive was necessary to enhance hepatitis C care at the Service, and the facilitators, challenges and successes of the project, and what professional learnings they gained throughout the project.” [Lines 176–182, Methods]

We are also able to include the interview guide in the supplementary material if the reviewer feels this would improve the paper.

We have addressed the comment about a coding framework in the next response.

  1. Reviewer: Page 5 line 194- It is unclear how NVIVO was used to develop coding frameworks or what the frameworks being referred to are. Were transcriptions of interviews coded in NVIVO?

Response: We have added the following text to the methods to outline the coding approach:

“The initial coding framework was developed deductively based on the semi-structured interview question guide, and was reviewed to identify emerging themes using an inductive approach.” [Lines 200–203, Methods]

  1. Reviewer: Page 5 line 212-84 were antibody positive but 54 were known antibody positive and received RNA testing. Based on study methods section, those individuals would not have been antibody tested? Were all 84-antibody tested through the study?

Response: Clients who were known to be antibody positive received an RNA test as their first test in the hepatitis C cascade of care. We realise that there was an issue with tense in the description of the results of the project and have changed the language to be clearer, so it now includes:

“Among the 84 clients who tested were hepatitis C antibody positive, 54 (64%; 95% CI: 90%–99%) were known (self-reported or Service clinical record) to be previously hepatitis C antibody positive and were hepatitis C RNA tested upon referral. Of the 84 clients who tested were hepatitis C antibody positive and eligible for hepatitis C RNA testing, three were lost to follow-up and did not receive a hepatitis C RNA test.”

[Lines 226–232, Results]

  1. Reviewer: The choice of an 11-month period pre intervention and a comparison period seems odd. Why not a 1-year period pre intervention? Please provide the rationale for the choice of 11 months

Response: We had access to 12 months of data to use as our historical comparator. However, the final month of this data overlapped with the first month of the intervention. Since we were unable to obtain any additional data, our historical comparator included 11 months of data.

  1. Reviewer: The fact that this was a single site study in one setting should also be listed as a limitation as this may not reflect the experience in other settings

Response: We have updated the limitations with the following text:

“Fifth, the project was implemented within a single sexual health service. The findings from this study may therefore not be applicable to services in other geographic and health settings.” [Lines 375–377, Limitations]

Reviewer 2 Report

Comments and Suggestions for Authors

This manuscript is clearly well written addressing effectiveness of an innovative model of HCV care among people who inject drugs applying mixed methods methodology. The following points are suggested with a hope for further improving the manuscript.

1.       I wonder if any reinfections had been tested and detected. Could a similar financial incentive increase post-SVR HCV testing for re-infection?

2.       In a similar context, I wonder if any uncured clients had been retreated, and if financial incentive could facilitate retreatment.

3.       How was the completion of treatment defined and ascertained? How about SVR?

4.       What kind of DAA was used for treatment? Who paid for the DAA treatment if a client was not covered by any insurance?

5.       Table 1 could also include columns for HCV-RNA positive (N=28) and negative (N=93) clients along with a column for p-values for comparisons between them.

Author Response

Reviewer two

This manuscript is clearly well written addressing effectiveness of an innovative model of HCV care among people who inject drugs applying mixed methods methodology. The following points are suggested with a hope for further improving the manuscript.

  1. Reviewer: I wonder if any reinfections had been tested and detected. Could a similar financial incentive increase post-SVR HCV testing for re-infection?

Response: There were no reinfections detected during the project implementation. We agree with the reviewer that offering financial incentives to enhance post-SVR testing for re-infection is an interesting and relevant proposal. However, it was outside the scope of this project, so we have not made any changes to the manuscript.

  1. Reviewer: In a similar context, I wonder if any uncured clients had been retreated, and if financial incentive could facilitate retreatment.

Response: The project did not systematically collect information on whether clients had been previously treated for hepatitis C, so we have not presented data on prior HCV treatment history.

  1. Reviewer: How was the completion of treatment defined and ascertained? How about SVR?

Response: We describe how completion of treatment was defined in lines 125-127 with the following text: “Collection of the first box of DAA treatment indicated treatment commencement, and collection of the final box of DAA treatment indicated treatment completion.”

We have added more information to this text to make it clearer for the reader how these were defined and ascertained:

“Collection of the first box of DAA treatment from the service pharmacy indicated treatment commencement, and collection of the final box of DAA treatment from the service pharmacy indicated treatment completion.” [Lines 189–193, Methods]

  1. Reviewer: What kind of DAA was used for treatment? Who paid for the DAA treatment if a client was not covered by any insurance?

Response: The project prescribed regimens of Glecaprevir/Pibrentasvir, Sofosbuvir/Velpatasvir or Ledipasvir/Sofosbuvir, depending on whether there was evidence of liver damage or cirrhosis. All clients who participated in the project were eligible for Australia’s universal health care insurance scheme (Medicare), so the need to identify another way to pay for DAA treatment was not required.

  1. Reviewer: Table 1 could also include columns for HCV-RNA positive (N=28) and negative (N=93) clients along with a column for p-values for comparisons between them.

Response: We have added a column for RNA positive clients. However, given our relatively small sample size, we did not conduct any statistical tests, as these would not have the statistical power to detect differences between groups. We have updated Table 1, and added the following text to the results section:

“Among the 28 clients who tested hepatitis C RNA positive, most were male (75%), and most did not identify Aboriginal and/or Torres Strait Islander (82%). Around half were aged between 35 and 49 years (43%) and had not previously attended the Service (50%).” [Results, lines 221–224]

Reviewer 3 Report

Comments and Suggestions for Authors

In the search for the dawn of hepatitis C as a public health problem, checking the efficacy of different initiatives for this goal is quite interesting.

Author Response

Reviewer three

Reviewer: In the search for the dawn of hepatitis C as a public health problem, checking the efficacy of different initiatives for this goal is quite interesting.

Response: We thank the reviewer for this comment and their review.

Round 2

Reviewer 1 Report

Comments and Suggestions for Authors

Thank you for addressing all concerns and comments. A final minor suggestion is for the abstract to increase clarity. Consider including the second sentence after the first phrase and then starting a new sentence explaining how the financial incentives were embedded ie. Between March 2020 and May 2021, financial incentives were embedded into as established person-centered hepatitis C model of care. Clients of the Cairns Sexual....

Page 3 line 67, consider use of term proposed instead of discussed. financial incentives have been "proposed" as an intervention...

Author Response

We would like to again thank the reviewer for their time reviewing our manuscript. 

  1. A final minor suggestion is for the abstract to increase clarity. Consider including the second sentence after the first phrase and then starting a new sentence explaining how the financial incentives were embedded ie. Between March 2020 and May 2021, financial incentives were embedded into as established person-centered hepatitis C model of care. Clients of the Cairns Sexual....

Response: We have updated the abstract to now include the following text:

“Between March 2020 and May 2021, financial incentives were embedded into an established person-centred hepatitis C model of care at Cairns Sexual Health Service. Clients of the service who self-reported experience of injecting drug use were offered an AU$20 cash incentive for hepatitis C testing, treatment initiation, treatment completion and test for cure.” [Lines 32–35, Abstract]

  1. Reviewer: Page 3 line 67, consider use of term proposed instead of discussed. financial incentives have been "proposed" as an intervention...

Response: We have updated the introduction to now include the following text:

“Providing financial incentives has been discussed proposed as an intervention which can overcome barriers to hepatitis C care, in particular economic and financial barriers.” [Lines 67–68, Introduction]